# Enhanced Cadmium Adsorption Dynamics in Water and Soil by Polystyrene Microplastics and Biochar

**DOI:** 10.3390/nano14131067

**Published:** 2024-06-21

**Authors:** Mengmeng Wang, Xuyou Jiang, Zhangdong Wei, Lin Wang, Jiashu Song, Peitong Cen

**Affiliations:** 1Miami College, Henan University, Kaifeng 475004, China; wmm0613@henu.edu.cn (M.W.); jiangxuyou2022@163.com (X.J.); a15603723229@163.com (J.S.); cenpeitong@163.com (P.C.); 2College of Geography and Environmental Science, Henan University, Kaifeng 475004, China

**Keywords:** microplastics, polystyrene, biochar, combined pollution, heavy metal speciation

## Abstract

Microplastics (MPs) are prevalent emerging pollutants in soil environments, acting as carriers for other contaminants and facilitating combined pollution along with toxic metals like cadmium (Cd). This interaction increases toxic effects and poses substantial threats to ecosystems and human health. The objective of this study was to investigate the hydrodynamic adsorption of Cd by conducting experiments where polystyrene microplastics (PS) and biochar (BC) coexisted across various particle sizes (10 µm, 20 µm, and 30 µm). Then, soil incubation experiments were set up under conditions of combined pollution, involving various concentrations (0.5 g·kg^−1^, 5 g·kg^−1^, 50 g·kg^−1^) and particle sizes of PS and BC to assess their synergistic effects on the soil environment. The results suggest that the pseudo-second-order kinetic model (*R*^2^ = 0.8642) provides a better description of the adsorption dynamics of Cd by PS and BC compared to the pseudo-first-order kinetic model (*R*^2^ = 0.7711), with an adsorption saturation time of 400 min. The Cd adsorption process in the presence of PS and BC is more accurately modeled using the Freundlich isotherm (*R*^2^ > 0.98), indicating the predominance of multilayer physical adsorption. The coexistence of 10 µm and 20 µm PS particles with BC enhanced Cd absorption, while 30 µm PS particles had an inhibitory effect. In soil incubation experiments, variations in PS particle size increased the exchangeable Cd speciation by 99.52% and decreased the residual speciation by 18.59%. The addition of microplastics notably impacted the exchangeable Cd speciation (*p* < 0.05), with smaller PS particles leading to more significant increases in the exchangeable content—showing respective increments of 45.90%, 106.96%, and 145.69%. This study contributes to a deeper understanding of the mitigation mechanisms of biochar in the face of combined pollution from microplastics and heavy metals, offering theoretical support and valuable insights for managing such contamination scenarios.

## 1. Introduction 

Microplastics, identified as emerging soil contaminants, are plastic fragments smaller than 5 mm in diameter [1,2,3,4]. They are mainly composed of synthetic organic polymers like polystyrene (PS), polyethylene (PE), and polypropylene (PP) [5], known for their cost-effectiveness and lightweight properties. When microplastics are exposed to the environment, they undergo physical, chemical, and biological weathering processes, leading to their fragmentation into tiny particles. These microplastic particles enter the soil through various pathways, including dry and wet deposition, surface runoff, wastewater discharge, the use of organic fertilizers, and plastic mulching films [6,7]. Consequently, they pose significant ecological risks to terrestrial ecosystems [8,9,10]. Recent studies have confirmed the presence of diverse microplastics in agricultural soil, with concentrations reaching 78 N·kg^−1^ in topsoil and 62.5 N·kg^−1^ in subsoil, respectively [11,12,13]. Due to their large specific surface area and numerous surface charges, microplastics have a high capacity to bind contaminants. They effectively adsorb both organic compounds and metal pollutants onto their surfaces, acting as vectors for these contaminants [14,15,16,17]. The interactions between microplastics and other pollutants lead to combined pollution that alters the environmental behavior of individual contaminants, posing distinct risks to biotic communities [18]. Particularly noteworthy are the synergistic effects of combined pollution involving cadmium and microplastics, known as Cd-MPs, which often result in more severe toxic impacts than those caused by either cadmium or microplastics alone. In natural environments, the interactions between microplastics and other toxic chemicals are influenced by various factors, such as the characteristics of microplastics (size, concentration, surface morphology, polymer type) [19] and environmental parameters (organic matter content, pH value, ion concentration) [20]. Consequently, predicting the pollution caused by microplastics, especially when combined with other pollutants, becomes increasingly challenging.

Cadmium is a major concern due to its persistence, high toxicity, mobility, and significant health risks [21]. It can enter the human body through air, water, soil, and food, accumulating primarily in organs like the kidneys, liver, and bones, leading to irreversible damage and potential diseases such as liver and kidney cancers [22]. A study on heavy metal pollution in Chinese agricultural fields revealed that 16.67% of farmland was contaminated to varying degrees, with cadmium constituting 25.20% of the contamination among eight surveyed heavy metals, making it the most prevalent metal contaminant in Chinese soils [23]. It is crucial to closely monitor the combined contamination of Cd and microplastics in soil environments to reduce their negative impacts on human health and ecological balance. Studies have shown that incorporating PS with low concentrations of cadmium (2 mg·L^−1^) can diminish the bioavailability of Cd, thereby alleviating its toxicity [24]. Moreover, interactions between ultra-low Cd concentrations (0.01 mg·L^−1^) and microplastics have been observed to lower the toxicity of heavy metals [25], leading to increased diversity and abundance of microbial communities. Remarkably, PE and polyvinyl chloride (PVC) exhibit substantial adsorption capacities for cadmium [26]. Nevertheless, the coexistence of microplastics and heavy metals is not always advantageous. The presence of high Cd concentrations with microplastics can intensify the toxicity of Cd [19,27], resulting in combined pollution. Furthermore, microplastics in aquatic environments have been found to modify the speciation of heavy metals [28], potentially heightening their biotoxicity.

Biochar, a sustainable material derived from agricultural waste and biomass through thermochemical transformation under anaerobic conditions, stands out as an excellent environmental remediation agent [29]. Its distinctive features include an extensive specific surface area, a robust carbon–nitrogen ratio, a comprehensive porous structure, and relatively stable functional groups [30,31]. Owing to its exceptional adsorption and catalytic properties, biochar has found widespread application in soil remediation and managing environmental pollution [32]. The adsorption of Cd by biochar reduces both its bioavailability and mobility. Moreover, biodegraded biochar demonstrates enhanced adsorptive capacity for Cd [33,34]. Numerous studies have indicated that the adsorption efficiency of biochar for heavy metals is influenced by factors such as the feedstock used, pyrolysis conditions, electrokinetic potential, types of functional groups, and metal speciation [35,36,37,38,39,40,41]. Via composite treatment, biochar can develop unique molecular structures that not only enhance its physicochemical properties [42] but also facilitate the formation of stable complexes or precipitates with Cd [43,44,45]. Furthermore, biochar can undergo chemical modifications to enhance its ion exchange capacity and specific surface area [46,47], thus improving its adsorption capabilities. Beyond its role in adsorbing heavy metals from the environment, biochar offers additional benefits such as enhancing agricultural physiology, mitigating the impacts of salinity on plant growth and development, and promoting animal growth [21]. This versatile amendment boasts a wide array of applications, making it a valuable tool for various environmental and agricultural purposes.

In recent years, extensive research has been conducted to investigate the combined pollution and interactions between microplastics and heavy metals in soil environments. Yu et al. demonstrated that microplastics can impact the levels of heavy metals by influencing physicochemical and biological geochemical processes in soil [48]. Moreover, microplastics can alter the soil’s physical and chemical properties, affecting the migration and transformation of heavy metal speciation, ultimately reducing the bioavailability of heavy metals. Studies confirmed that the adsorption of Cd by microplastics is rapid initially, with smaller microplastic particles demonstrating superior adsorption capacity for Cd [49]. Specifically, PE particles ranging from 100–154 μm exhibited a maximum adsorption capacity of 30.5 μg·g^−1^, with the adsorbed Cd easily desorbed from the microplastics. While current research predominantly focuses on the removal and remediation of individual pollutants by biochar, particularly in aquatic systems, there exists a significant gap in understanding the adsorptive properties of Cd in soil environments where both microplastics and biochar are present. Further exploration in this area is crucial for comprehensive environmental remediation strategies. The research on the forms and concentration changes in Cd has significant shortcomings. Therefore, this study focuses on Cd as the target pollutant. Cd solutions of different concentrations (0, 10, 20, and 50 mmol·L^−1^) were prepared using cadmium nitrate (Cd(NO_3_)_2_·4H_2_O) to simulate cadmium-polluted environments. Biochar and microplastics were added to these cadmium solutions to investigate whether the competitive adsorption of cadmium by microplastics affects the adsorption efficiency of biochar. The physicochemical properties of biochar and microplastics were characterized and analyzed to evaluate their impact on adsorption efficiency. Furthermore, by altering the particle size and concentration in the soil, the impact of PS microplastics on the Cd form was analyzed. The primary goal of this study is to offer essential theoretical support for devising environmental protection strategies and guiding agricultural development.

## 2. Materials and Methods

### 2.1. Competitive Adsorption of Biochar

#### 2.1.1. Adsorption Kinetics Experiments

The polystyrene microplastics (PS) utilized were procured from Huachuang Plastic Raw Materials in Zhangmutou, Dongguan City, Guangdong Province, with three distinct particle sizes selected: 10 µm, 20 µm, and 30 µm, designated as PS10, PS20, and PS30, respectively. The biochar (BC) was obtained from Henan Institute of Science and Technology. All materials were of guaranteed reagent grade. 

Polystyrene with the smallest particle size, PS10, was used to create a composite environment of PS-Cd. This study aimed to investigate the competitive adsorption of biochar on cadmium over a 24-h period. Each treatment was conducted in triplicate. The experimental data were modeled using both the pseudo-first-order and pseudo-second-order kinetic models, expressed as follows:

Pseudo-first-order kinetic (PFOM) model:(1)$$\frac{dq_{t}}{dt}=k_{l}\left(q_{e}-q_{t}\right)$$

Pseudo-second-order kinetic (PSOM) model:(2)$$\frac{dq_{t}}{dt}=k_{2}{\left(q_{e}-q_{t}\right)}^{2}$$
where *q_e_* represents the equilibrium adsorption capacity, expressed in g·kg^−1^; *q_t_* is the adsorption capacity at time t, also in g·kg^−1^; *k*_1_ is the rate constant of the pseudo-first-order kinetics, in h^−1^; and *k*_2_ is the rate constant of the pseudo-second-order kinetics, in h^−1^.

#### 2.1.2. Adsorption Isotherm Experiments

This study included four distinct experimental treatments: CK (control, with BC) and biochar supplemented with polystyrene microplastics of different sizes—PS10 (PS10+BC), PS20 (PS20+BC), and PS30 (PS30+BC). The specific ratios and designations for the components within these composite systems are detailed in Table 1. Each treatment was replicated three times to ensure reproducibility. All chemical reagents used were guaranteed reagent grade and standardized before use to ensure the reliability of the experiments. To quantify the adsorptive interaction of biochar with Cd, both the Freundlich and Langmuir isotherm models were utilized. The corresponding equations are as follows:

Freundlich isotherm model:(3)$$q_{e}=K_{f}C_{e}^{n}$$

Freundlich isotherm model:(4)$$q_{e}=\frac{K_{L}Q_{max}C_{e}}{1+K_{1}C_{e}}$$
where *Ce* represents the equilibrium concentration of cadmium in the solution, measured in mg·L^−1^; *q_e_* denotes the equilibrium adsorption capacity of the particles, in mg·kg^−1^; *K_f_* is the Freundlich adsorption capacity coefficient, expressed as g^(1−n)^·L^n^·kg^−1^; *K_L_* is the Langmuir adsorption capacity coefficient, in L·mg^−1^; *n* is the adsorption intensity constant; and *Q_max_* represents the maximum adsorption capacity.

#### 2.1.3. Soil Incubation Experiment under PS and Cd Co-Existence

The soil used in this study was collected from the central ground surface of a landfill situated in the eastern outskirts of Kaifeng City, Henan Province, China (34.76° N, 114.39° E). The sampled soil belongs to the fluvo-aquic category and was obtained from the arable layer (0–20 cm). The soil was naturally air-dried and cleared of any extraneous materials like stones and plant roots after collection. The physico-chemical properties of the soil, as detailed in Table 2, are based on the previous study by Wang, L. et al. [50]. 

Soil samples were air dried and sifted through a 0.25 mm sieve. For each experiment, 100 g of soil were placed into a conical flask and mixed with 50 mL of ultrapure water to achieve a consistent mud-like texture. Polystyrene microplastics (PS), selected for their various sizes (10 µm, 20 µm, 30 µm), were added in different concentrations: 0.5 g·kg^−1^ (C1), 5 g·kg^−1^ (C2), and 50 g·kg^−1^ (C3). After a 30-day incubation, the soil samples were dried in an oven at 105 °C for two hours. Throughout the soil incubation experiments, all conditions, including temperature (25 °C), humidity (55%), and pH (5.0), were strictly controlled to ensure consistency of the results. The weights of the soil samples were then accurately recorded post drying. The specific details of the experimental conditions can be found in Table 3.

The polystyrene microplastics (PS) utilized were procured from Huachuang Plastic Raw Materials in Zhangmutou, Dongguan City, Guangdong Province, with three distinct particle sizes selected: 10 µm, 20 µm, and 30 µm, designated as PS10, PS20, and PS30, respectively. The biochar (BC) was obtained from Henan Institute of Science and Technology.

### 2.2. Environmental Analysis

#### 2.2.1. Characterization Analysis of Test Materials

This study established a composite adsorption system that includes multiple pollutants, such as polystyrene microplastics (PS) and the heavy metal cadmium (Cd). Detailed investigations on the adsorption kinetics and isotherm were conducted through batch experiments. 

The surface area, pore size, and pore volume of the materials were quantified using the V-Sorb 2800P surface area and pore size analyzer (Beijing Haiguang, Beijing, China). Fourier-transform infrared spectroscopy (FTIR) from Bruker (Ettlingen, Germany) was employed to characterize the functional groups present on the surfaces of both polystyrene microplastics and biochar.

#### 2.2.2. Adsorption Kinetics Experiments

For this experiment, 0.125 g of both 10 µm polystyrene microplastics (PS) and biochar (BC) were individually weighed and placed into separate 250 mL conical flasks. Afterwards, 200 mL of a 20 mmol·L^−1^ cadmium nitrate solution was added to each flask, with the pH adjusted to 5.0. Adsorption kinetics were observed over a 24-h period at specific time intervals: 0 min, 5 min, 10 min, 15 min, 30 min, 1 h, 2 h, 4 h, 8 h, 16 h, and 24 h. The flasks were sealed and incubated in an orbital shaker at 25 °C under dark conditions, with an agitation speed of 150 rpm. 

To analyze the Cd(II) concentration, 5 mL samples were periodically taken from the supernatant, filtered through a 0.22 µm microporous membrane, and analyzed using a flame atomic absorption spectrophotometer (AAS) (Model GGX-830, Beijing Haiguang, China). The average values obtained from the different treatments were used for further analysis.

#### 2.2.3. Adsorption Isotherm Experiments

A weight of 0.025 g polystyrene (PS) and 0.025 g biochar (BC) were combined and then added to 200 mL of cadmium nitrate solutions at concentrations of 0, 10, 20, and 50 mmol·L^−1^, respectively. The mixtures were transferred into separate 250 mL conical flasks, which were sealed and gently agitated to ensure a homogeneous solution. These flasks were then incubated at a constant temperature of 25 °C, with agitation maintained at 150 rpm for a period of 24 h. After the incubation period, a syringe was used to withdraw 5 mL of the supernatant from each flask. The supernatant was filtered through a 0.22 µm microporous membrane, and the concentration of Cd(II) (*Ce*) in the filtrate was measured. The average values obtained from each treatment were calculated for further analysis. 

#### 2.2.4. Speciation Analysis of Test Cadmium

The speciation of cadmium (Cd) in soil samples was analyzed using an advanced BCR sequential extraction procedure based on the method of Rauret, G. [51]. This method involves a systematic extraction process that separates Cd into its exchangeable, reducible, oxidizable, and residual forms. After centrifugation and filtration, the extracted fractions were stored at 4 °C for further analysis. The quantification of cadmium content in each fraction obtained through the BCR method was performed using AAS (Atomic Absorption Spectrometry). The AAS was calibrated with standard solutions before each experiment to ensure measurement accuracy.

### 2.3. Data Analysis

Data processing and statistical analyses were conducted using Microsoft Office Excel 2019 (Microsoft Corp., Redmond, WA, USA). Graphical representations were generated with Origin 2019b (OriginLab, Northampton, MA, USA). One-way ANOVA and Pearson correlation analysis were used to determine the statistical significance of the differences through SPSS 26.0 (IBM Corp., Armonk, NY, USA). The results are presented as mean ± standard deviation (mean ± SD). Significant differences among treatments at the 5% level were denoted by distinct lowercase letters and analyzed using SPSS. The Duncan multiple range test was employed to assess significant differences between treatments, with a significance threshold set at *p* < 0.05.

## 3. Results and Discussion

### 3.1. Analysis of Material Properties

#### 3.1.1. Pore Size Analysis of PS and BC

The results of the pore size analysis for polystyrene (PS) and biochar (BC) were presented in Table 4. Both BC and PS10 exhibited similar specific surface areas, the surface areas were ranked as follows: PS20 > PS10 > PS30. Remarkably, PS20 demonstrated a larger specific surface area of 15.055 m^2^·g^−1^, which contradicted the expected theoretical outcome. This anomaly challenges the theory that smaller particles should have larger specific surface areas. It is possible that the aging of microplastics had contributed to this phenomenon, as it could have created more adsorption sites [52]. Consequently, the increased surface area may have enhanced the adsorption of metals or organic pollutants.

Biochar exhibited a wide range of pore sizes, ranging from 2.3 nm to 288.7 nm, indicating a significant porous structure. In terms of pore volume, the microplastics were ranked as follows: PS10 > PS20 > PS30, with PS30 displaying the smallest average pore volume at 0.0218 cm^3^·g^−1^. However, PS30 also displayed the broadest range of pore sizes (2.02–334.9 nm), suggesting a complex pore architecture that potentially offered greater adsorption capabilities. Correlation analysis revealed a negative association between the specific surface area and the pore structure in relation to particle size, with a correlation coefficient *R*^2^ = 0.568.

Based on the adsorption–desorption curves depicted in Figure 1, it is evident that all four particle types exhibit Type IV-a behavior, characterized by Type H3 hysteresis loops. This suggests that they possess a mesoporous structure with comparable unfilled pore sizes, providing ample adsorption sites for the target pollutant, Cd.

#### 3.1.2. Functional Group Analysis of PS and BC

The functional groups present in microplastics and biochar play a crucial role in determining their properties, as evidenced by the infrared spectroscopy results depicted in Figure 2. While both materials exhibit similar types of functional groups, there are noticeable differences in the intensities of these groups. Specifically, BC exhibits significantly stronger intensities of alcohols and phenolic hydroxyl groups (O-H) within the 3650–3580 cm^−1^ range compared to polystyrene (PS). Additionally, unsaturated C-H bonds are prominent above 3000 cm^−1^ in biochar, whereas polystyrene displays enhanced intensities of unsaturated double bonds =C-H and aromatic C-H in the 3010–3040 cm^−1^ range. This disparity may explain the relatively weaker adsorption capacity of biochar, as noted in previous studies [26]. 

In the 2000–1600 cm^−1^ range, which corresponds to the C=O and C=C stretching vibrations, biochar displayed significantly higher absorption intensity compared to polystyrene. These organic functional groups, such as C=O, play a vital role in adsorbing, retaining, and removing excess bioavailable cadmium from solutions and cadmium-contaminated soils [36], thereby enhancing biochar’s effectiveness in cadmium adsorption. At 1600 cm^−1^, PS s exhibited an absorption peak characteristic of benzene rings. Research suggests that Cd(II) ion adsorption on PS can occur through a cation-π mechanism involving neutral or low-charge functional groups, a phenomenon that is more pronounced in highly carbonized biochar. This mechanism may contribute more over threefold to the adsorption capacity compared to ion-exchange capabilities, potentially leading to significant PS-Cd combined pollution [37]. Lastly, in the region between 1300 and 1000 cm^−1^ corresponding to single-bond C-O vibrations, both materials exhibited consistent profiles.

### 3.2. Adsorption Characteristics of BC for Cd in Coexistence with PS

#### 3.2.1. Adsorption Kinetics Analysis

The adsorption kinetics of Cd(II) by PS and BC are presented in Figure 3. As the adsorption time progresses, the rate of Cd(II) adsorption by both PS and BC initially increases rapidly, followed by a gradual decrease, eventually reaching an equilibrium state. The equilibrium time for Cd(II) adsorption is approximately 400 min, with the adsorption capacity ranking as PS > BC. In the initial stage of adsorption, both materials efficiently adsorb Cd(II) from the solution through rapid surface adsorption, primarily driven by physical adsorption on the particles’ surfaces [39]. PS exhibits a significantly higher rate of adsorption compared to BC, which can be attributed to its superior specific surface area and pore structure, particularly PS10. The second stage involves a longer duration of adsorption, characterized by various chemical adsorption mechanisms such as surface complexation, ion exchange, chelation, electrostatic attraction, inner-sphere complexation, and redox reactions. After approximately 400 min, a stable equilibrium is reached, where no further changes in the adsorption capacity are observed.

To investigate the adsorption kinetics of Cd(II) from solutions by PS and BC, this study employed the pseudo-first-order kinetics equation (PFOM) and the pseudo-second-order kinetics equation (PSOM) to fit the experimental data. The optimal fitting parameters obtained are presented in Table 5. Some disparities were observed between the theoretical adsorption capacities and the actual adsorption capacities for both models. In general, a lower *k* value indicates a faster reaction rate [17], and it was found that the *k* value for PSOM model was smaller than that for the PFOM model. This suggests that chemical adsorption predominates over physical adsorption on the biochar surface. Moreover, the correlation coefficients *R*^2^ for the PSOM model were higher, at 0.8642 and 0.8008, compared to those for the PFOM model, which were 0.7771 and 0.7137, respectively. Therefore, the pseudo-second-order kinetics model (PSOM) provides a more accurate description of the adsorption dynamics of microplastics and biochar for cadmium in this study. These findings indicate that the adsorption equilibrium in the composite system is reached more promptly, highlighting the significant influence of competitive adsorption of Cd by PS on the efficiency of cadmium adsorption by biochar.

#### 3.2.2. Adsorption Isotherm Models

In the study, we characterized the adsorption of Cd(II) from solutions using two isotherm models: the Langmuir and Freundlich models. The adsorbents employed were PS10 and BC. Figure 4 illustrates the variations in adsorption capacity relative to the equilibrium concentration.

The optimized fitting parameters for different isotherm adsorption models can be observed in Table 6. The adsorption of Cd(II) by PS10 and BC closely follows the Freundlich model, as indicated by the higher correlation coefficients *R*^2^ of 0.9889 and 0.9288, respectively, compared to those of the Langmuir model, which are 0.9159 and 0.8019. This suggests that the predominant adsorption mechanism involves multilayer physical adsorption on the surface [53]. The adsorption isotherms for Cd(II) by PS demonstrate strong agreement with the Freundlich model, consistent with findings reported by Holmes in [54]. Therefore, this study adopts the Freundlich isotherm model to characterize the adsorption of cadmium by microplastics and biochar.

#### 3.2.3. Cadmium Adsorption by PS and BC

The adsorption isotherms for cadmium by biochar in conjunction with different particle sizes of polystyrene (PS) are depicted in Figure 5. The adsorption capacity follows the following ranking: PS30 > PS20+BC > PS10+BC > PS30+BC > PS20 > PS10. The data demonstrate that the combination of biochar with smaller PS particles (PS10 and PS20) enhances cadmium adsorption compared to adsorption by microplastics alone. This enhancement can be attributed partly to biochar’s smaller specific surface area relative to PS10 and PS20, along with its superior ion exchange capabilities [55], which contribute to its overall adsorption capacity. As a result, biochar outperforms PS in cadmium uptake when they coexist. Furthermore, biochar’s unique structural features, specifically its bonding with Cd(II) to electron-rich domains similar to graphene structures, significantly enhance Cd(II) adsorption. This mechanism can have an impact three times greater than the quantitative contribution of biochar’s ion exchange capacity [37]. Additionally, the diverse organic functional groups inherent in biochar also augment its adsorption efficacy. However, the synergy of biochar with PS30 slightly inhibits Cd(II) adsorption, likely due to the specific surface area and complex pore architecture of PS30. BET data confirm that despite having a smaller surface area than biochar, PS30 features a broader range of pore sizes and a more elaborate pore structure, which facilitates more effective Cd(II) adsorption.

Table 7 illustrates the fitting parameters of the Freundlich model for the adsorption of cadmium by biochar in conjunction with different particle sizes of polystyrene (PS). Notable differences were observed in the adsorption constant (*n*), with the following order: PS30 > PS10 > PS20. 

Regarding the Freundlich adsorption capacity coefficient (*K_f_*), the ranking is as follows: PS30 > PS20 > PS10, with PS10 showing a significantly larger difference compared to the other two sizes. When PS was combined with BC, the *K_f_* value for PS10 was substantially reduced compared to the PS10+BC combination, while for PS30, the *K_f_* value slightly exceeded that of PS30+BC. These findings support the experimental results indicating that the combination of biochar and PS30 partially inhibits the adsorption of Cd. Furthermore, the correlation coefficients (*R*^2^) for the Freundlich model of cadmium adsorption in different composite treatments involving PS and biochar consistently exceeded 0.9, demonstrating robust model fits.

### 3.3. Cd Speciation Changes in Soil with Coexisting PS and BC

#### 3.3.1. Effects of PS on the Content and Change Rate of Cd Speciation

Table 8 provides detailed information regarding the content of various cadmium speciation forms when PS of different particle sizes and concentrations coexist with BC. The findings reveal that the particle size and concentration of microplastics predominantly exert a significant positive influence on the exchangeable speciation of cadmium. Additionally, they have both positive and negative effects on reducible and oxidizable speciations, while only minimally affecting the residual speciation.

In the control (CK) treatment, the concentration of exchangeable cadmium was 0.91 mg·kg^−1^. The addition of microplastics resulted in a significant increase in this speciation, with the PS10C2 treatment showing the highest rise at 158.28%, while the PS30C3 treatment displayed the least increase at 36.26%. For reducible cadmium, the initial concentration under CK was 1.35 mg·kg^−1^. It slightly decreased under the PS10 treatment but increased under the PS20 and PS30 treatments. Notably, the PS20C1 and PS20C3 treatments exhibited increases of 48.89% and 52.59%, respectively. Oxidizable cadmium had a baseline concentration of 0.32 mg·kg^−1^ under CK, with negligible changes observed across treatments. However, the PS10C2 treatment increased the oxidizable cadmium concentration to 0.35 mg·kg^−1^, representing a 26.52% increase compared to CK. Under CK conditions, the residual cadmium concentration was 3.51 mg·kg^−1^. In all other treatments, there was a decrease in residual cadmium concentration.

#### 3.3.2. Effects of PS on Total Cd Content and Change Rate of Cd Speciation

For CK, no significant changes were observed compared to the initial conditions assessed 30 days prior. The recovery rate of cadmium remained consistent, ranging between 96.85% and 111.69%. The total cadmium content varied from 6.28 mg·kg^−1^ in the direct extraction group to a range of 6.09 to 7.02 mg·kg^−1^ in the other treatments. Overall, there was minimal fluctuation in the change in total cadmium content, with decreases ranging from 0.08% to increases of up to 15.24%. The presence of polystyrene (PS) of various sizes and concentrations did not have a significant impact on the total content of cadmium. 

#### 3.3.3. Distribution of Cd Speciation in Soil with Coexisting PS and BC

Figure 6 effectively illustrates the proportions of various cadmium speciations in soil when PS coexists with BC. The figure demonstrates a noticeable increase in the content of exchangeable cadmium, while the oxidizable and reducible speciations show minimal changes. Additionally, the residual speciation exhibits a decrease. The impact of microplastics on different cadmium speciations revealed significant variations (*p* < 0.05) in the exchangeable, reducible, and oxidizable speciations. The exchangeable cadmium showed increases ranging from +89% to +158%, the reducible form varied from −13% to +48%, and the oxidizable form fluctuated between −66% and +26%. Conversely, the changes in the residual form were not statistically significant (*p* > 0.05), with only a minimal decrease observed ranging from −0.8% to −0.3%. These results underscore the significant role of microplastics in altering cadmium speciation in soil. Further comparative analyses reveal statistically significant differences in cadmium speciation between the experimental setup and CK.

#### 3.3.4. Effects of Different Particle Sizes of PS on Cd Speciation

When combined with biochar, the content of various cadmium speciations under different particle size microplastic treatments is depicted in Figure 7. Figure 7A, it is evident that as the particle size of polystyrene (PS) increases, the enhancement in exchangeable cadmium content proportionally decreases. Specifically, increments of 145.69%, 106.96%, and 45.90% are observed for PS sizes of 10 µm, 20 µm, and 30 µm, respectively. Figure 7B illustrates that PS microplastics with a particle size of 10 μm reduce the content of reducible cadmium, whereas sizes of 20 μm and 30 μm lead to an increase. The influence of PS microplastics on oxidizable cadmium is shown in Figure 7C. It demonstrates that 10 μm PS microplastics exert minimal influence on oxidizable cadmium content, while sizes of 20 μm and 30 μm result in a reduction of oxidizable cadmium content. Finally, Figure 7D reveals that all tested particle sizes of PS decrease the content of residual cadmium, with a 20 μm size exhibiting the most significant reduction of 22.89%.

The influence of particle size of polystyrene (PS) microplastics on the contents of exchangeable, reducible, oxidizable, and residual cadmium in soil highlights the significant impact of particle size on the exchangeable and residual speciation of cadmium (*p* < 0.01). Therefore, when combined with biochar, the particle size of PS microplastics plays a crucial role in determining cadmium speciation in soil.

#### 3.3.5. Effects of Different Concentrations of PS on Cd speciation

When combined with biochar, the content of various cadmium speciations under different concentration microplastic treatments is presented in Figure 8. In Figure 8A, it is evident that the exchangeable cadmium content consistently increases across different microplastic concentrations, showing a significant enhancement compared to the control group. Figure 8B, demonstrates a positive correlation between the concentration of polystyrene microplastics and the content of reducible cadmium, with a notable peak observed at 50 g·kg^−1^ treatment. The impact on oxidizable cadmium is depicted in Figure 8C, where minimal influence is observed at the lowest concentration, but significant reductions are noted at higher concentrations. Lastly, Figure 8D illustrates a consistent decline in residual cadmium content across all PS concentrations, highlighting the extent to which each concentration influences the residual speciation.

While there are no significant changes observed in the exchangeable, reducible, or oxidizable speciations of cadmium, the concentration of polystyrene microplastics does significantly impact the residual form (*p* < 0.05). This finding underscores the critical role of microplastic concentration in modifying the speciation of cadmium when biochar is also present.

### 3.4. Mechanism of Biochar in the Co-Pollution of PS Microplastics and Cd

Microplastics with three different particle sizes (10 μm, 20 μm, 30 μm) and biochar exhibit mesoporous structures. Generally, the specific surface area decreases as the particle size increases; however, the 20 μm PS microplastic shows the largest specific surface area. Larger particles possess more complex pore structures, which likely contribute to their enhanced adsorption capacity compared to biochar. The increased complexity in pore structure is crucial as it provides additional sites for cadmium attachment, potentially enhancing the overall adsorption capacity. Additionally, the hierarchical pore distribution in larger microplastic particles may facilitate faster diffusion rates of cadmium ions, thereby improving adsorption kinetics. These factors together suggest that microplastics with larger, more intricate pore structures could play a pivotal role in cadmium immobilization. This finding is consistent with the results of Zhao et al. [56], which also demonstrated the significant role of pore structure in enhancing cadmium adsorption. Both materials exhibit similar types of functional groups, but there are notable differences in the intensity of unsaturated C-H and C=O bonds. Biochar’s higher unsaturated C-H bond intensity enhances cadmium attachment through π-π interactions, while the stronger C=O bonds in microplastics favor cadmium retention via complexation reactions. This complementary interaction improves the overall adsorption of cadmium.

The adsorption dynamics of cadmium on PS and biochar follow pseudo-second-order kinetics rather than first-order kinetics, indicating that an adsorption equilibrium time of 400 min is required. Furthermore, the adsorption behavior of PS and biochar for cadmium aligns closely with the Freundlich adsorption model (*R*^2^ > 0.98), suggesting that multilayer physical adsorption predominates in the cadmium uptake of both materials. The combination of biochar with PS enhances cadmium adsorption. However, the rich pore structure of 30 μm PS microplastics may result in stronger adsorption compared to biochar alone, thereby exerting a suppressive effect on the adsorptive performance of biochar in their coexistence. This result demonstrates competitive adsorption dynamics when both microplastics and biochar are present. Therefore, optimizing the ratio of biochar to microplastics is crucial for maximizing adsorption efficiency.

The influence of microplastics with different sizes and concentrations on the speciation of cadmium indicates that higher concentrations and smaller sizes of polystyrene (PS) microplastics have more noticeable effects on the various forms of cadmium. The introduction of microplastics has a significant impact on the exchangeable state of cadmium, which increases as the particle size decreases. The exchangeable state is the most biotoxic and mobile form of cadmium. Therefore, the presence of microplastics exacerbates the environmental and health risks associated with this form of cadmium. 

In recent years, biochar has emerged as a prominent research focus in the fields of environmental remediation and pollutant removal due to its low cost, rich porous structure, and diverse functional groups. Ren et al. investigated the synergistic effects of biochar and microplastics on the removal of organic compounds in agricultural settings. Their study revealed that the combination of biochar and microplastics significantly enhances the removal efficiency of organic compounds from soil [57]. Furthermore, Liu et al. achieved a substantial improvement in the adsorption of heavy metal ions, specifically cadmium (Cd(II)) and lead (Pb(II)), in aquatic environments by optimizing the pore structure and surface functional groups of both biochar and modified biochar [58]. These results are consistent with the competitive adsorption outcomes for biochar and microplastics observed in aqueous environments in this study. Additionally, this research focused on the effects of biochar application on cadmium speciation in soils subjected to composite pollution. The findings revealed that the presence of microplastics significantly increased the content of exchangeable cadmium in the soil (*p* < 0.05), which agrees with the results reported by Miao et al. [59].

Meanwhile, both biochar and microplastics undergo aging in the natural environment, which may affect the remediation efficacy of biochar. Therefore, future research should focus on systematically evaluating the long-term adsorption performance and passivation effects of biochar on cadmium under conditions of combined pollution. It is particularly important to assess the effectiveness and stability of biochar under varying pollution concentrations and soil types to enhance its potential application in real-world agricultural settings.

## 4. Conclusions

The adsorption–desorption isotherms of the selected biochar and polystyrene microplastics in this study exhibit Type IV-a classification, with H3 hysteresis loops, indicating predominantly mesoporous structures. A clear correlation was observed, where smaller particle diameters corresponded to increased specific surface areas. It is worth noting that PS10 had the smallest average particle diameter and pore volume, while PS30 exhibited a more intricate pore architecture, contributing to its enhanced adsorption capacity for the heavy metal Cd(II). Although there were minimal variations in the functional groups among the different particle types, significant differences were evident in the intensity of unsaturated C-H and C=O bonds. These bonds play a crucial role in adsorption processes.

The results of the adsorption kinetics indicated that the pseudo-second-order kinetic model (*R*^2^ = 0.8642) was more suitable for describing the adsorption dynamics of cadmium by the selected biochar and polystyrene microplastics. The adsorption process initially occurred rapidly, primarily through physical adsorption, followed by a slower phase of chemical adsorption that reached equilibrium at 400 min. 

The results of the adsorption isotherms demonstrated that the adsorption of cadmium by both biochar and polystyrene microplastics was consistent with the Freundlich isotherm model (*R*^2^ > 0.98). This finding indicates that multilayer physical adsorption predominates in the adsorption of cadmium by both materials.

The investigation into the effects of microplastics with varying particle sizes and concentrations on cadmium speciation, in combination with biochar, revealed significant impacts. Specifically, the particle size of microplastics had a significant effect on the exchangeable state of cadmium, resulting in an average increase of 99.52% (*p* < 0.05). On the other hand, significant reductions were observed in the residual state of cadmium, with an average decrease of 18.59% (*p* < 0.01). Furthermore, the concentration of microplastics significantly altered the content of residual cadmium (*p* < 0.05), while no significant differences were noted in the exchangeable, reducible, and oxidizable states of cadmium.

The co-presence of polystyrene microplastics and cadmium amplifies the complexity of soil pollution. The interaction between PS and cadmium significantly enhances the exchangeable and available speciations of cadmium, thereby increasing its mobility in the soil. Additionally, this interaction synergistically enhances the effectiveness of biochar in adsorbing cadmium. Given its adaptability and cost effectiveness, biochar emerges as an exceptionally efficient passivating agent for remediating soil contaminated with the PS-Cd composite. This offers distinct advantages in environmental cleanup efforts, highlighting the potential of biochar for addressing soil pollution issues.

## Figures and Tables

**Figure 1 nanomaterials-14-01067-f001:**
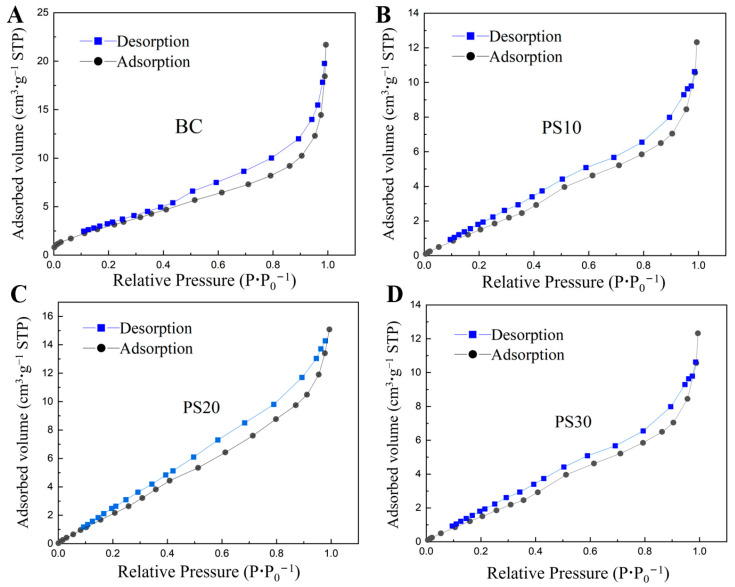
Adsorption–desorption curves for: (**A**) BC; (**B**) PS10; (**C**) PS20; and (**D**) PS30.

**Figure 2 nanomaterials-14-01067-f002:**
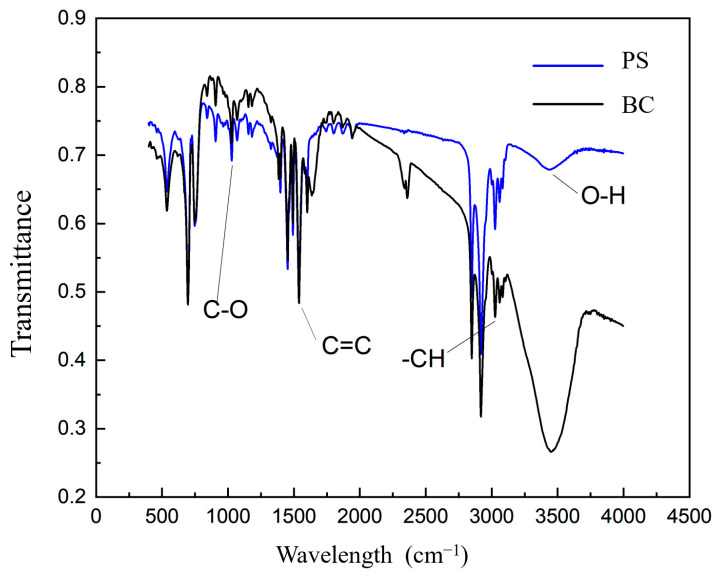
Infrared spectroscopy graphs of BC, SPS, MPS, and LPS.

**Figure 3 nanomaterials-14-01067-f003:**
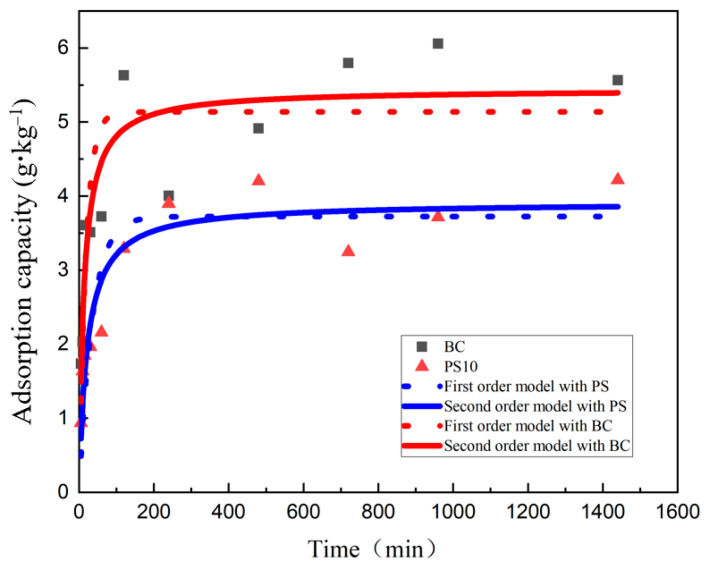
Adsorption kinetics and fitting.

**Figure 4 nanomaterials-14-01067-f004:**
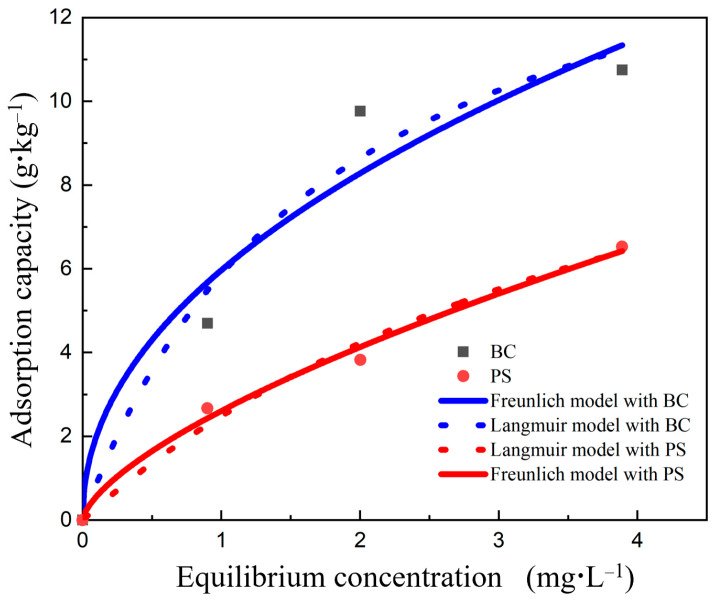
Isotherms and fitting results for Cd(II) adsorption by PS and BC.

**Figure 5 nanomaterials-14-01067-f005:**
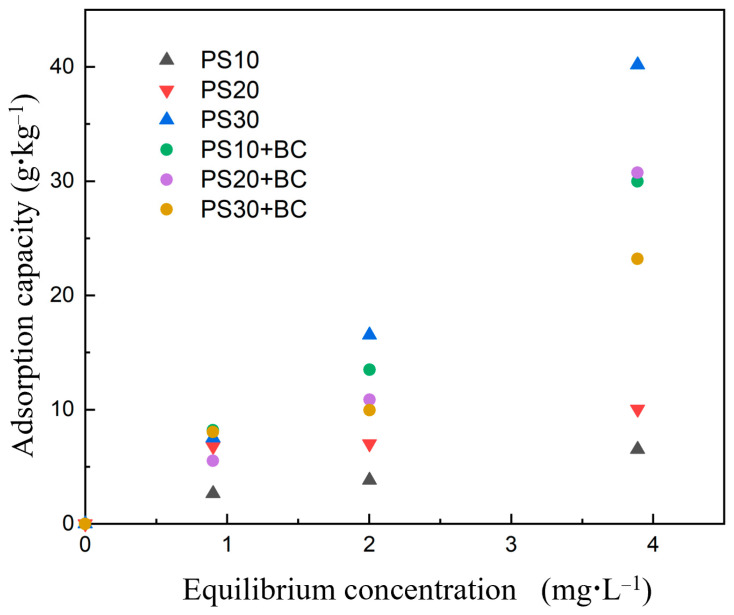
Adsorption isotherms of biochar in the coexistence of cadmium and polystyrene microplastics of different sizes.

**Figure 6 nanomaterials-14-01067-f006:**
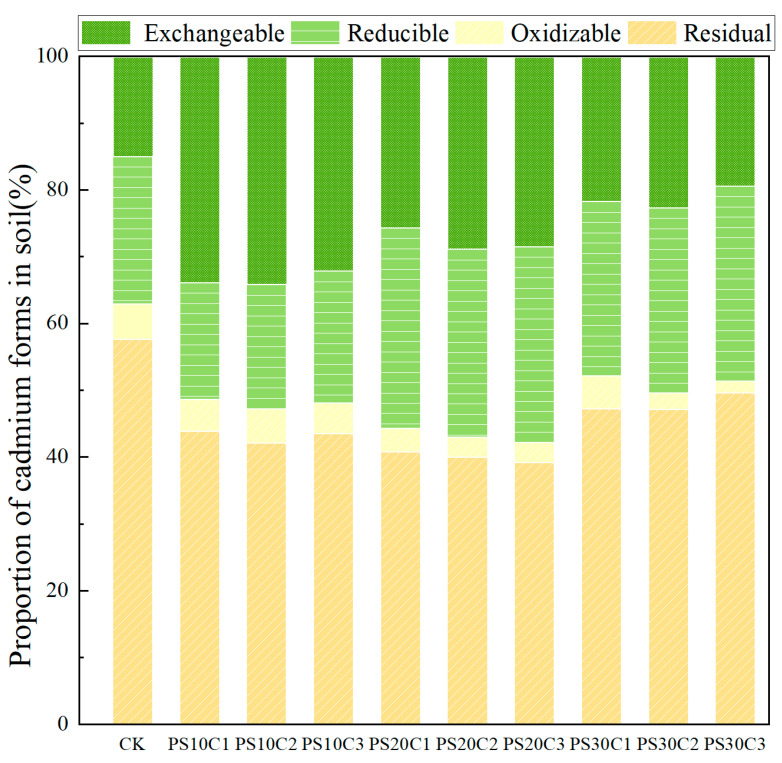
Proportions of cadmium speciation in soil with coexisting PS and BC.

**Figure 7 nanomaterials-14-01067-f007:**
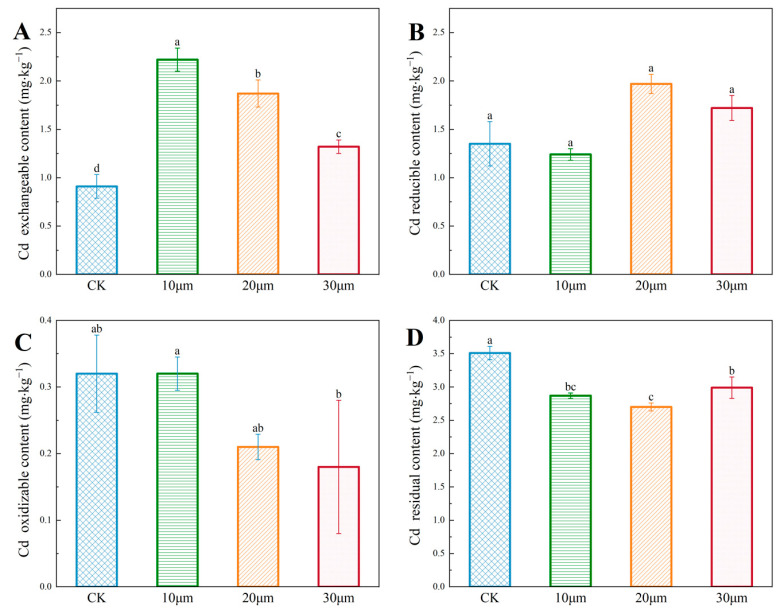
Cadmium speciation in soil under different particle sizes of PS in combined pollution: (**A**) Cd exchangeable content; (**B**) Cd reducible content; (**C**) Cd oxidizable content; (**D**) Cd residual content. Different lowercase letters indicate significant differences (*p* < 0.05).

**Figure 8 nanomaterials-14-01067-f008:**
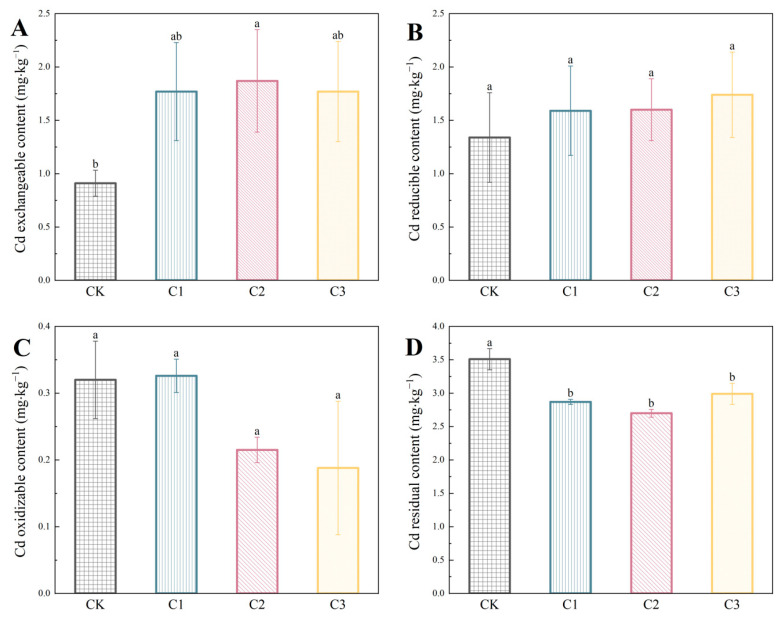
Cadmium speciation in soil under different concentrations of PS in combined pollution: (**A**) Cd exchangeable content; (**B**) Cd reducible content; (**C**) Cd oxidizable content; (**D**) Cd residual content. Different lowercase letters indicate significant differences (*p* < 0.05).

**Table 1 nanomaterials-14-01067-t001:** The specific ratio and naming method of BC and PS.

Naming	BC	PS10 (10 μm)	PS20 (20 μm)	PS30 (30 μm)
BC	1	0	0	0
PS10+BC	1	1	0	0
PS20+BC	1	0	1	0
PS30+BC	1	0	0	1

**Table 2 nanomaterials-14-01067-t002:** Physico-chemical properties of the soil samples from the study area.

	Indicators	Mean ± SD
Property	pH	7.68 ± 0.36
	OM (%)	2.03% ± 0.95%
	TN (%)	16% ± 4%
	AP (g·kg^−1^)	0.24 ± 0.05
Element	Cr (mg·kg^−1^)	90.96 ± 11.01
	Ni (mg·kg^−1^)	252.31 ± 24.08
	Cu (mg·kg^−1^)	128.58 ± 25.36
	Zn (mg·kg^−1^)	486.57 ± 105.31
	As (mg·kg^−1^)	15.82 ± 0.83
	Cd (mg·kg^−1^)	6.09 ± 0.16
	Pb (mg·kg^−1^)	80.57 ± 7.94

**Table 3 nanomaterials-14-01067-t003:** Experimental conditions.

Experiment No.	PS Addition (g)	PS Particle Size (μm)	Water (mL)	Soil Sample (g)	Soil Microplastic Content (g·kg^−1^)
CK	0	/	50	100	/
PS10C1	0.05	10	50	100	0.5
PS10C2	0.5	10	50	100	5
PS10C3	5	10	50	100	50
PS20C1	0.05	20	50	100	0.5
PS20C2	0.5	20	50	100	5
PS20C3	5	20	50	100	50
PS30C1	0.05	30	50	100	0.5
PS30C2	0.5	30	50	100	5
PS30C3	5	30	50	100	50

Note: The “/” symbol indicates that no microplastics were added.

**Table 4 nanomaterials-14-01067-t004:** Specific surface area, pore volume, and pore width of PS and BC.

Particles	Specific Surface Area (m^2^·g^−1^)	Pore Volume (cm^3^·g^−1^)	Pore Size and Range (nm)
BC	12.899	0.0346	10.307 (2.3–288.7)
PS10	13.694	0.0308	7.667 (2.2–208.8)
PS20	15.055	0.0271	6.237 (2.2–301.7)
PS30	9.689	0.0218	7.297 (2.02–334.9)

**Table 5 nanomaterials-14-01067-t005:** Optimal fitting parameters for different kinetic models of Cd(II) adsorption by PS10 and BC.

Model	Parameter	PS10	BC
First-order $\log\left(q_{e}-q_{t}\right)$ $=\log q_{e}-k_{1}t$	*q_e_* theoretical (g·kg^−1^)	3.7235	5.1361
	*q_e_* actual (g·kg^−1^)	4.2167	5.8065
	*k*_1_ (h^−1^)	0.0282	0.0541
	*R* ^2^	0.7771	0.7137
Second-order $\frac{t}{q_{t}}=\frac{1}{k_{2}q_{e}^{2}}+\frac{1}{q_{e}}t$	*q_e_* theoretical (g·kg^−1^)	2.8257	5.4431
	*q_e_* actual (g·kg^−1^)	4.2167	5.8065
	*k*_2_ (kg·g^−1^h^−1^)	0.0116	0.0139
	*R* ^2^	0.8642	0.8008

Note: “*q_e_*” represents the equilibrium adsorption capacity, while “*q_t_*” refers to the adsorption capacity at time *t*. The first-order adsorption rate constant is denoted as “*k*_1_” and the second-order adsorption rate constant is represented by “*k*_2_”.

**Table 6 nanomaterials-14-01067-t006:** Optimal fitting parameters for different isotherm models of Cd(II) adsorption by PS and BC.

Model	Parameter	PS10	BC
Langmuir	*Qmax* (g·kg^−1^)	14.132	16.392
	*K_L_* (L·g^−1^)	0.213	0.558
	*R^2^*	0.9159	0.8019
Freundlich	*N*	0.664	0.473
	*K_2_*(g^(1−n)^·L^n^·kg^−1^)	2.605	5.963
	*R^2^*	0.9889	0.9288

**Table 7 nanomaterials-14-01067-t007:** Freundlich model fitting parameters for Cd adsorption by microplastics.

Adsorbent Material	*n*	*K_f_*	*R* ^2^
PS10	0.664	2.605	0.9889
PS20	0.296	6.456	0.9649
PS30	1.249	7.327	0.9966
PS10+BC	1.022	7.381	0.9848
PS20+BC	1.400	4.552	0.9890
PS30+BC	0.917	6.479	0.9439

**Table 8 nanomaterials-14-01067-t008:** Content and rate of Cd speciation in soil with coexisting PS and BC.

Group	Exchangeable Content (mg·kg^−1^)	Rate (%)	Reducible Content (mg·kg^−1^)	Rate (%)	Oxidizable Content (mg·kg^−1^)	Rate (%)	Residual Content (mg·kg^−1^)	Rate (%)
CK	0.91 ± 0.122 d	/	1.35 ± 0.229 ef	/	0.32 ± 0.058 abc	/	3.51 ± 0.158 ab	/
PS10C1	2.25 ± 0.324 a	+147.73	1.16 ± 0.115 g	−14.07	0.32 ± 0.024 ab	+0.09	2.91 ± 0.201 a	−17.09
PS10C2	2.34 ± 0.549 a	+158.28	1.27 ± 0.156 fg	−5.92	0.35 ± 0.077 a	+26.52	2.88 ± 0.314 b	−17.95
PS10C3	2.10 ± 0.001 ab	+131.06	1.29 ± 0.317 ef	−4.44	0.30 ± 0.046 abc	−5.14	2.84 ± 0.025 ab	−19.09
PS20C1	1.72 ± 0.123 bc	+89.48	2.01 ± 0.335 abc	+48.89	0.24 ± 0.026 abc	−8.80	2.73 ± 0.253 ab	−22.22
PS20C2	1.91 ± 0.631 a	+111.01	1.86 ± 0.095 ab	+37.78	0.20 ± 0.020 abc	−37.76	2.64 ± 0.309 ab	−24.79
PS20C3	2.00 ± 0.224 abc	+120.40	2.06 ± 0.419 a	+52.59	0.21 ± 0.117 bc	−53.93	2.75 ± 0.564 ab	−21.65
PS30C1	1.35 ± 0.304 cd	+49.07	1.62 ± 0.004 cd	+20.00	0.31 ± 0.023 abc	−6.64	2.93 ± 0.230 ab	−16.52
PS30C2	1.38 ± 0.061 cd	+147.73	1.69 ± 0.125 de	+25.19	0.15 ± 0.011 abc	−35.95	2.87 ± 0.177 ab	−18.23
PS30C3	1.24 ± 0.426 cd	+36.26	1.87 ± 0.077 bcd	+38.52	0.11 ± 0.037 c	−66.77	3.17 ± 0.285 ab	−9.69

Note: Different lowercase letters indicate significant differences (*p* < 0.05).

## Data Availability

Data is contained within the article.

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
