# Peer review of "Enhanced Cadmium Adsorption Dynamics in Water and Soil by Polystyrene Microplastics and Biochar"

_nanomaterials, 2024, doi:10.3390/nano14131067_

Round 1

Reviewer 1 Report

Comments and Suggestions for Authors

The article presented is well described and covers a topic of great interest in the world's scientific community. However, I have doubts about some vitally important aspects:
1) Hasn't quality control been carried out? It is mentioned very briefly and I think it should be given more space in the document.
2) What are the quality grades and markings of the elements used in this research?
3) Can the results obtained in other fields be relevant?

Author Response

Dear reviewer,

Thank you for your decision and constructive comments on my manuscript. We have carefully considered the suggestion of Revisions and make changes. We have tried our best to improve and made some changes in the manuscript (3042434). The grey part that has been revised according to your comments. Revisions notes, point-to-point, are given as follows:

Revision 1#

Hasn't quality control been carried out? It is mentioned very briefly and I think it should be given more space in the document.

Response: Thank you for pointing this out. We have expanded the description of quality control measures in our manuscript to ensure clarity and transparency. Specifically, we have included the following details:

  1. Reagent and material quality: All reagents and materials were of guaranteed reagent (GR) grade to ensure the accuracy of our experiments. Detailed information about each material and reagent can be found in Table 1.
  2. Instrument Calibration: All analytical instruments have been calibrated before use and regularly maintained during the experimental process to ensure the accuracy of measurement data. For example, the flame atomic absorption spectrophotometer (AAS) was calibrated with standard solutions before each experiment to ensure measurement accuracy. This regular calibration ensures the reliability of our Cd(II) concentration measurements.
  3. Experimental Repetition: Each treatment was repeated three times to ensure data reliability. This replication allows for statistical analysis and validation of our results, providing a robust basis for our conclusions.
  4. Control experimental conditions: In order to ensure the consistency of experimental conditions and the reliability of results, we strictly controlled the experimental conditions including temperature, humidity, and pH.
  5. Data processing: Rigorous data processing techniques were applied to ensure the integrity and accuracy of the results. All data were processed using statistical software, and outliers were identified and handled appropriately. The mean values and standard deviations were calculated for each set of triplicate measurements. Pearson correlation analysis and one-way ANOVA were conducted to assess the significance of the differences between treatments.

Revised:

  • Line 134: All materials were of guaranteed reagent grade.
  • Line 137: Each treatment was conducted in triplicate.
  • Line 151-153: Each treatment was replicated three times to ensure reproducibility. All chemical reagents used were guaranteed reagent grade and standardized before use to ensure the reliability of the experiments.
  • Line 180-182: Throughout the soil incubation experiments, all conditions, including temperature (25℃), humidity(55%), and pH(5.0), were strictly controlled to ensure consistency of the results.
  • Line 234-235: The AAS was calibrated with standard solutions before each experiment to ensure measurement accuracy.

Revision 2#

What are the quality grades and markings of the elements used in this research?

Response: Thank you for your interest in the quality grades and markings of the materials used in our research. All materials utilized in this study underwent strict quality control measures, and their quality grades and markings were meticulously recorded to ensure the reliability and reproducibility of our results. Below are the specific details of the key materials we used:

Table 1 Specifications and Sources of Materials and Reagents Used

Material

Chemical Formula

Quality Grade

Source

Microplastics

Guaranteed Reagent

Zhangmutou Huachuang Plastic Raw Materials Store, Dongguan City, Guangdong Province, China (10 μm, 20 μm, 30 μm) purity >99%

Biochar

Guaranteed Reagent

Henan Institute of Science and Technology, China (produced from corn stover, pyrolyzed at 500°C)

Cadmium Nitrate Solution

Cd(NO₃)â‚‚·4Hâ‚‚O

Guaranteed Reagent

Sigma-Aldrich, USA

Acetic Acid

CH₃COOH

Guaranteed Reagent

Tianjin Kermel Chemical Reagent Co., Ltd., China

Hydroxylamine Hydrochloride

NH3OH·HCl

Guaranteed Reagent

Shanghai Macklin Biochemical Technology Co., Ltd., China

30% Hydrogen Peroxide

Hâ‚‚Oâ‚‚

Guaranteed Reagent

Yantai Shuangshuang Chemical Co., Ltd., China

Ammonium Acetate

CH₃COONH₄

Guaranteed Reagent

Tianjin Kermel Chemical Reagent Co., Ltd., China

Hydrofluoric Acid

HF

Guaranteed Reagent

Tianjin Kermel Chemical Reagent Co., Ltd., China

Perchloric Acid

HClOâ‚„

Guaranteed Reagent

Tianjin Xinyuan Chemical Co., Ltd., China

Revision 3#

Can the results obtained in other fields be relevant?

Response: Thank you for this insightful question. Biochar has emerged as a prominent research focus in the fields of environmental remediation and pollutant removal due to its low cost, rich porous structure, and diverse functional groups. Therefore, biochar is often used as a passivation agent to address environmental pollution problems. In actual soil environments, compound pollution is frequently encountered. Consequently, our research places a significant emphasis on the combined pollution of microplastics and heavy metal cadmium.

In our manuscript, we have conducted a more in-depth analysis of the original results. Additionally, we have cited the research findings of several scholars in different fields regarding biochar, microplastics, and cadmium speciation, which align with our results. This cros- referencing not only confirms our research results in other research fields, but also provides us with new research ideas and possible application directions.

Revised:

  • Line 484-492: The increased complexity in pore structure is crucial as it provides additional sites for cadmium attachment, potentially enhancing the overall adsorption capacity. Additionally, the hierarchical pore distribution in larger microplastic particles may facilitate faster diffusion rates of cadmium ions, thereby improving adsorption kinetics. These factors together suggest that microplastics with larger, more intricate pore structures could play a pivotal role in cadmium immobilization. This finding is consistent with the results of Zhao et al [58], which also demonstrated the significant role of pore structure in enhancing cadmium adsorption.
  • Line 493-496: Biochar's higher unsaturated C-H bond intensity enhances cadmium attachment through π-π interactions, while the stronger C=O bonds in microplastics favor cadmium retention via complexation reactions. This complementary interaction improves the overall adsorption of cadmium.
  • Line 505-508: This result demonstrates competitive adsorption dynamics when both microplastics and biochar are present. Therefore, optimizing the ratio of biochar to microplastics is crucial for maximizing adsorption efficiency.
  • Line 516-537: In recent years, biochar has emerged as a prominent research focus in the fields of environmental remediation and pollutant removal due to its low cost, rich porous structure, and diverse functional groups. Ren et al. investigated the synergistic effects of biochar and microplastics on the removal of organic compounds in agricultural settings. Their study revealed that the combination of biochar and microplastics significantly enhances the removal efficiency of organic compounds from soil [59]. Furthermore, Liu et al. achieved a substantial improvement in the adsorption of heavy metal ions, specifically cadmium (Cd(II)) and lead (Pb(II)), in aquatic environments by optimizing the pore structure and surface functional groups of both biochar and modified biochar [60]. These results are consistent with the competitive adsorption outcomes for biochar and microplastics observed in aqueous environments in this study. Additionally, this research focused on the effects of biochar application on cadmium speciation in soils subjected to composite pollution. The findings revealed that the presence of microplastics significantly increased the content of exchangeable cadmium in the soil (P<0.05), which is in agreement with the results reported by Miao et al [61].

Meanwhile, both biochar and microplastics undergo aging in the natural environment, which may affect the remediation efficacy of biochar. Therefore, future research should focus on systematically evaluating the long-term adsorption performance and passivation effects of biochar on cadmium under conditions of combined pollution. It is particularly important to assess the effectiveness and stability of biochar under varying pollution concentrations and soil types to enhance its potential application in real-world agricultural settings.

We appreciate your insightful suggestions, which have significantly contributed to improving the quality and clarity of our manuscript. We believe the revised version now addresses all the issues raised and hope it meets your expectations.

Thank you once again for your time and effort in reviewing our manuscript. We look forward to your feedback on the revised version.

Best regards,

Name: Zhangdong Wei

Address: Henan University Miami College, Kaifeng, China 475004

E-Mail: [email protected]

Reviewer 2 Report

Comments and Suggestions for Authors

nanomaterials-3042434

In this manuscript, a study on the role of biochar in experiments with combined pollution from microplastics and heavy metals, specifically cadmium, was carried out. Soil environments polluted with cadmium solutions of different concentrations were prepared to examine the physicochemical properties of biochar and microplastics. The competitive adsorption effect of microplastics on the cadmium adsorption efficiency of biochar was also investigated. Moreover, changing the particle size and concentration in the soil, the impact of polystyrene microplastics on the cadmium form was analyzed. Results evidenced that structural features of biochar significantly enhance cadmium adsorption.

Interesting results have emerged from comparison of polystyrene and biochar, highlighting different features during cadmium adsorption.

The manuscipt is interesting and it describes an important aspect coupling the use of waste or biomass to produce biochar, the latter evidencing a high affinity to cadmium, in presence of microplastics.

Revisions

line 46: ‘wlarge’ change to ‘large’;

line 98: ‘I This’ change to ‘This’;

line 117: ‘Cd solutions of different concentrations were prepared …’ which was the salt of Cd used to prepare Cd solutions? Please specify;

lines 118-119: ‘Biochar and microplastics were introduced to examine the physicochemical properties of biochar and microplastics.’ Please, check the sentence;

line 131: ‘The biochar (BC) was obtained from Henan Insitute of Science and Technology.’ Which was the native material used to obtain biochar? Are in literature reported possible different behaviors of biochar depending on its origin?;

line 162: In this study, the fluvo-aquic category of soil, obtained from the top 0-20 cm surface layer, was used. In the Conclusions section of this manuscript, it should be reported that soils can show differences and a high number of variables and the model proposed in this study should be adapted, accordingly.  

line 164: ‘gelongs’ change to ‘belongs’;

line 199: ’24-hours period’: Is the adsorption of Cd maintained for long period, allowing metals recovery from the substrate and their elimination from soils?;

line 209: ‘0.025’ at the beginning of the sentence it would be better to avoid numbers.

Author Response

Dear reviewer,

Thank you for your decision and constructive comments on my manuscript. We have carefully considered the suggestion of Revisions and make changes. We have tried our best to improve and made some changes in the manuscript(3042434). The yellow part that has been revised according to your comments. Revisions notes, point-to-point, are given as follows:

Revision 1#

‘wlarge’ change to ‘large’

Response:

line 46: We have revised the word in the article. 

Revision 2#

‘I This’ change to ‘This’

Response:

line 98: We have revised the word in the article.

Revision 3#

‘Cd solutions of different concentrations were prepared …’ which was the salt of Cd used to prepare Cd solutions? Please specify;

Response: Thank you for your suggestion. We have revised the manuscript to specify the cadmium salt (Cd(NO₃)₂) used to prepare the Cd solutions for clarity and completeness.

Original: Cd solutions of different concentrations were prepared to simulate cadmium-polluted environments.

Revised:

lines 117-119: Cd solutions of different concentrations (0, 10, 20, and 50 μmol·L−1) were prepared using cadmium nitrate (Cd(NO₃)â‚‚·4Hâ‚‚O) to simulate cadmium-polluted environments.

Revision 4#

‘Biochar and microplastics were introduced to examine the physicochemical properties of biochar and microplastics.’ Please, check the sentence.

Response: Thank you for pointing out the need for clarity in this sentence. The original sentence aimed to explain the introduction of biochar and microplastics into cadmium solutions to investigate their competitive adsorption behavior and to characterize their physicochemical properties. However, we realize that the wording was not clear enough. To improve clarity and ensure logical flow, we have revised the sentence.

Original: Biochar and microplastics were introduced to examine the physicochemical properties of biochar and microplastics. The study also investigates the competitive adsorption effect of microplastics on the Cd adsorption efficiency of biochar.

Revised:

line 119-122: Biochar and microplastics were added to these cadmium solutions to investigate whether the competitive adsorption of cadmium by microplastics affects the adsorption efficiency of biochar. The physicochemical properties of biochar and microplastics were characterized and analyzed to evaluate their impact on adsorption efficiency.

Revision 5#

‘The biochar (BC) was obtained from Henan Insitute of Science and Technology.’ Which was the native material used to obtain biochar? Are in literature reported possible different behaviors of biochar depending on its origin?

Response:

The biochar (BC) used in this study was obtained from the Henan Institute of Science and Technology and was produced from corn stover.

Research has shown that the properties of biochar can vary significantly depending on the feedstock and pyrolysis conditions. [1,2]

Corn stover is often used to produce biochar because it is a plentiful and renewable resource, low-cost, and easy to obtain. It contains a high amount of organic carbon, which, after pyrolysis, converts into high-carbon-content biochar with a porous structure. These characteristics give it excellent adsorption properties. Additionally, using corn stover to produce biochar helps reduce agricultural waste, increase soil organic matter, and enhance plant growth and agricultural productivity. It also sequesters carbon, reducing greenhouse gas emissions, thus offering significant environmental benefits. Therefore, biochar produced from corn stover shows great potential as an amendment for remediating environmental pollution.

  1. Kalina, M.; Sovova, S.; Svec, J.; Trudicova, M.; Hajzler, J.; Kubikova, L.; Enev, V. The Effect of Pyrolysis Temperature and the Source Biomass on the Properties of Biochar Produced for the Agronomical Applications as the Soil Conditioner. Materials202215, 8855. https://doi.org/10.3390/ma15248855
  2. Rafiq, M.K.; Bachmann, R.T.; Rafiq, M.T.; Shang, Z.; Joseph, S.; Ansari, T.M. Influence of Pyrolysis Temperature on Physico-Chemical Properties of Corn Stover (Zea mays L.) Biochar and Feasibility for Carbon Capture and Energy Balance. PLOS ONE 2016, 11, e0156894. https://doi.org/10.1371/journal.pone.0156894

Revision 6#

In this study, the fluvo-aquic category of soil, obtained from the top 0-20 cm surface layer, was used. In the Conclusions section of this manuscript, it should be reported that soils can show differences and a high number of variables and the model proposed in this study should be adapted, accordingly. 

Response: In this study, we selected soil samples from the arable layer(0-20 cm) because this layer is the most impacted by agricultural activities. The frequent use of agricultural chemicals such as fertilizers and pesticides results in higher concentrations of heavy metal pollutants in this layer. Moreover, the arable layer is where plant roots are most densely distributed, and root activity significantly influences the soil's physicochemical properties and the mobility of heavy metals. Analyzing this soil layer provides more representative experimental data, enabling a more accurate assessment of pollution levels.

Original: The sampled soil belongs to the fluvo-aquic category and was obtained from the top 0-20 cm surface layer.

Revised:

line 170-171: The sampled soil belongs to the fluvo-aquic category and was obtained from the arable layer (0-20 cm).

Revision 7#

‘gelongs’ change to ‘belongs’

Response:

line 170: We have revised the word in the article.

Revision 8#

‘24-hours period’: Is the adsorption of Cd maintained for long period, allowing metals recovery from the substrate and their elimination from soils?

Response: Thank you for your valuable feedback. The study focuses on the adsorption of heavy metal cadmium by microplastics and biochar. We observed that within 24 hours, the adsorption reached its peak with no desorption occurring. Consequently, we selected a 24-hour period to capture this peak adsorption behavior. Furthermore, investigating how aged microplastics and biochar affect cadmium adsorption is a key area we plan to explore in future research.

Revision 9#

‘0.025’ at the beginning of the sentence it would be better to avoid numbers.

Response: Thank you for your suggestion. We have revised the sentence to avoid starting with a number, enhancing readability.

Original: "0.025 g of polystyrene (PS) and biochar (BC) were combined and then added to 200 mL of cadmium nitrate solutions at concentrations of 0, 10, 20, and 50 50mmolžL-1, respectively. "

Revised:

line 217-219:  "A sample of 0.025 g of polystyrene (PS) and biochar (BC) was combined and then added to 200 mL of cadmium nitrate solutions at concentrations of 0, 10, 20, and 50 mmolžL-1, respectively."

We believe these revisions have significantly improved the clarity and quality of our manuscript. We sincerely appreciate your constructive comments, which have been invaluable in enhancing our work. We hope that the revisions meet your expectations and address these concerns. Thank you for your time and consideration. We are looking forward to your feedback.

Best regards,

Name: Zhangdong Wei

Address: Henan University Miami College, Kaifeng, China 475004

E-Mail: [email protected]

Round 2

Reviewer 2 Report

Comments and Suggestions for Authors

nanomaterials-3042434-peer-review-v2

In the revised version of the manuscript, all the points raised during the review process have been taken into account by the authors and the manuscript has been modified, accordinlgly.

In particular, aspects and details related to the efficiency of biochar adsorption depending on biochar origins, the features of soil layer considered in this study for cadmium removal, and the maintenance of metals adsorption to biochar, have been specified.

This manuscript represents an interesting study and can give useful insights fo future research.